This is a Registered Report and may have an associated publication; please check the article page on the journal site for any related articles.

REGISTERED REPORT PROTOCOL

# Efficacy and safety of Yunkang oral liquid combined with conventional therapy for threatened miscarriage of first-trimester pregnancy a protocol for systematic review and meta-analysis

Peng Sun[1☯], Liping Tang[2☯], Dongmei Yan[2☯], Bin li[2☯], Lingxia Xu[2☯]*, Fei Wang[2☯]*

**1** The Affiliated Hospital of Jiangxi University of Chinese Medicine, Nanchang, Jiangxi, China, **2** Academician Workstation, Jiangxi University of Chinese Medicine, Nanchang, Jiangxi, China

☯ These authors contributed equally to this work.
* xulingxia94@163.com (LX); skywf@163.com (FW)

## Abstract

### Introduction

Threatened miscarriages is a common complication of first-trimester pregnancy. Due to the beneficial effects, there are increasing clinical studies on Yunkang oral liquid(YKOL). However, the efficacy and safety of YKOL are still unknown. The aim of this systematic review was to assess the efficacy and safety of YKOL in the treatment of threatened miscarriage during the first-trimester pregnancy (TMFP).

### Methods

This protocol will be prepared according to the preferred reporting items for systematic review and meta-analysis protocols (PRISMA-P) statement. The systematic review will include all randomized controlled trials (RCTs) studies published until April 2021. Electronic sources including CNKI, WF, VIP, CBM, MEDLINE(PubMed), Embase, Cochrane Library, and Web of Science will be searched for potentially eligible studies. The international clinical trial registration platform and the Chinese clinical trial registration platform of controlled trials will be searched from their inception until April 1st, 2021. According to the inclusion and exclusion criteria, screening literature, extraction data will be conducted by two researchers independently. Statistical analysis will use RevMan 5.3.5 software. The strength of evidence from the studies will be evaluated with the Grading of Recommendation, Assessment, Development and Evaluation (GRADE) methods.

### Results

This study will provide evidence for YKOL combined with conventional therapy for TMFP.

**Data Availability Statement:** No datasets were generated or analyzed during the current study. All relevant data from this study will be made available upon study completion.

**Funding:** This work is supported by National Natural Science Foundation of China (No. 81202911 and 81360509), Evidence based ability improvement project of traditional Chinese Medicine in The State Administration of Traditional Chinese Medicine of China (No.2019XZZX-LG005), Academician Workstation of Jiangxi University of Traditional Chinese Medicine Project (No. ysgzz201806) and First-class Discipline Construction Funding of Jiangxi Province (No. JXSYLXK-ZHYAO148). The funders had no role in study design, data collection and analysis, decision to publish, or preparation of the manuscript.

**Competing interests:** The authors have declaredthat no competing interests exist.

## Conclusion

The efficacy and safety of YKOL combined with conventional therapy for TMFP will be assessed.

## Systematic review registration

INPLASY202140105 (https://www.doi.org/10.37766/inplasy2021.4.0105).

## Background

Threatened miscarriage, as a common complication of first-trimester pregnancy, occurred in about 30% to 40% [1]. The clinical manifestations of threatened miscarriage include vaginal bleeding, abdominal pain, lumbago, a closed cervix and an intrauterine viable fetus [2]. Besides, in the early trimester of pregnancy, approximately half of threatened miscarriage patients suffer from miscarriage, which brings heavy medical and economic burdens to pregnant women and their families [3]. At present, the reason that causes threatened miscarriage is still unclear. Many studies have suggested that the primary reason of threatened miscarriage of first-trimester pregnancy (TMFP) principally involve embryonic chromosomal abnormality, low levels of circulating progesterone [4], Some infections, endocrine disorders [5], and immune system disorders [6]. Although many drugs were reported to be used for the treatment of TMFP, such as progesterone [7,8], β-human chorionic gonadotropin [9], magnesium sulfate [10], and phloroglucinol [10]. The treatment that target reducing the risk of TMFP with high effectiveness is unavailable. conventional therapy include standard supportive care, human chorionic gonadotropin, progestogen drugs and other placebo. However, conventional therapy cannot reduce the occur of abdominal pain and lumbago.

In China, Traditional Chinese Medicine (TCM) preparation has been widely used for the treatment of TMFP [11]. Due to the Two Child Policy in China, the number of pregnant woman is increasing. The incidence of risk of threatened miscarriage which is associated with adverse pregnant outcomes is also increasing. In pregnancy, The main effects of Yunkang oral liquid (YKOL) is mainly composed of 23 kinds of Chinese medicinal materials (Table 1) [12], are to promote maternal and child health, and alleviate the medical problems of pregnant women [13,14]. In addition, nither adverse effects nor toxicity of YKOL has been reported. YKOL is a TCM preparation made of *Dioscorea opposita Thunb.*, *Dipsacus asper Wall. Ex Henry*, *Astragalus membranaceus (Fisch.) Bge.*, *Angelica sinensis (Oliv.) Diels*, *Cibotium barometz (L.) J.Sm.*, *Cuscuta australis R.Br.*, *Taxillus chinensis (DC.) Danser*, *Eucommia ulmoides Oliv.*, *Psoralea corylifoliaL.*, *Codonopsis pilosula (Franch.)Nannf.*, *Poria cocos (Schw.) Wolf*, *Atractylodes macrocephala Koidz.*, *Equus asinus L.*, *Rehmannia glutinosa (Gaertn.) DC.*, *Cornus officinalis Sieb. et Zucc.*, *Lcycium barbarumL.*, *Prunus mume (Sieb.) Sieb.etZucc.*, *Paeonia lactiflora Pall.*, *Amomum villosum Lour.*, *Alpinia oxyphylla Miq.*, *Boehmeria nivea(L.)Gaud.*, *Scutellaria baicalensis Georgi*, *Artemisia argyi Levl.et Vant.* [12,15]. Animal studies have found that YKOL could effectively reduce the embryo loss rate in pregnant mice model [15,16]. This effect is related to increasing the hormone levels of follicle stimulating hormone, luteinizing hormone, prolactin, progesterone, and estrogenic in serum and promoting the proteins expression of prolactin receptor, progesterone receptor, and estrogenic receptor in mouse decidua [13–16]. In addition, animal studies also have found that YKOL may induce estrogen and progesterone receptors by phosphorylation via the classic Akt and Erk1/2 signaling pathways in the maternal-fetal interface of pregnant rats, thereby reducing the pregnancy loss rate

**Table 1. Basic information of YKOL.**

| Chinese botanical drugs | Latin name | Part of botanical drugs | Proportion |
|---|---|---|---|
| Shan yao | *Dioscorea opposita Thunb.* | rhizome | 125g |
| Xu duan | *Dipsacus asper Wall. Ex Henry* | root | 75g |
| Huang qi | *Astragalus membranaceus (Fisch.) Bge.* | root | 100g |
| Dang gui | *Angelica sinensis (Oliv.) Diels* | root | 75g |
| Gou ji(qumao) | *Cibotium barometz (L.) J.Sm.* | rhizome | 100g |
| Tu si zi(bing) | *Cuscuta australis R.Br.* | seed | 75g |
| Sang ji sheng | *Taxillus chinensis (DC.) Danser* | stalk | 50g |
| Du zhong(chao) | *Eucommia ulmoides Oliv.* | bark | 75g |
| Bu gu zhi | *Psoralea corylifoliaL.* | fruit | 75g |
| Dang shen | *Codonopsis pilosula (Franch.)Nannf.* | root | 75g |
| Fu ling | *Poria cocos (Schw.) Wolf* | sclerotium | 100g |
| Bai zhu(jiao) | *Atractylodes macrocephala Koidz.* | rhizome | 75g |
| e'jiao | *Equus asinus L.* | skin | 25g |
| Di huang | *Rehmannia glutinosa (Gaertn.) DC.* | rhizome | 100g |
| Shan zhu yu | *Cornus officinalis Sieb. et Zucc.* | fruit | 75g |
| Gou qi zi | *Lcycium barbarumL.* | fruit | 100g |
| Wu mei | *Prunus mume (Sieb.) Sieb.etZucc.* | fruit | 50g |
| Bai shao | *Paeonia lactiflora Pall.* | root | 75g |
| Sha ren | *Amomum villosum Lour.* | fruit | 50g |
| Yi zhi | *Alpinia oxyphylla Miq.* | fruit | 50g |
| Zhu ma gen | *Boehmeria nivea(L.)Gaud.* | root | 75g |
| Huang qin | *Scutellaria baicalensis Georgi* | root | 50g |
| Ai ye | *Artemisia argyi Levl.et Vant.* | foliage | 8.3g |

and improving the live birth rate [17]. The reported mechanisms from modern researches of YKOL various components are as follows: *Semen Cuscutae Chinensis* and *Dipsacales* are reported to have steroid-like and hormone-like effects, maintaining the hormone balance of the pregnant mother [15]; *Cibotii Rhizoma* maintain immune balance [18]; *Asini Corii Colla*, which has always been used as a tonic, promotes blood circulation [19].

Currently, the systematic reviews of YKOL have never been reported. Therefore, the purpose of this study was to explore the efficacy and safety of YKOL as adjuvant treatment for TMFP by pooling the current randomized controlled trials, in order to provide a high-quality clinical evidence.

## Methods

### Study registration

This systematic review protocol has been registered on INPLASY website and registration number were INPLASY202140105 (https://www.doi.org/10.37766/inplasy2021.4.0105). The protocol for this study was drafted according to the systematic review and meta-analysis preferred reporting project(PRISMA-P).

### Eligibility criteria

**Types of study.** The types of studies include parallel-design, cross-over design, but we only included the data at the end of the first stage for cross-over design. Case–control, cohort studies, Case series and studies without controls were excluded.

**Types of participants.** Participants must be individuals diagnosed with threatened miscarriage, and woman with gestational age between 1 and 12 weeks. Threatened miscarriage is diagnosed on the basis of documented fetal cardiac activity on ultrasound with a history of vaginal bleeding in the presence of a closed cervix [20]. There will be no restrictions based on age, race/ethnicity, socioeconomic status, orgeographic region.

**Types of interventions.** *Experimental interventions.* The experimental interventions include studies using Chinese Patent Medicine(YKOL) as a basic formula, regardless of the dose, method of dosing, or duration of administration, in combination with conventional therapy compared with conventional therapy excluding bed rest and psychological supports. YKOL is a TCM preparation made of *Dioscorea opposita Thunb.*, *Dipsacus asper Wall. Ex Henry*, *Astragalus membranaceus (Fisch.) Bge.*, *Angelica sinensis (Oliv.) Diels*, *Cibotium barometz (L.) J.Sm.*, *Cuscuta australis R.Br.*, *Taxillus chinensis (DC.) Danser*, *Eucommia ulmoides Oliv.*, *Psoralea corylifoliaL.*, *Codonopsis pilosula (Franch.)Nannf.*, *Poria cocos (Schw.) Wolf*, *Atractylodes macrocephala Koidz.*, *Equus asinus L.*, *Rehmannia glutinosa (Gaertn.) DC.*, *Cornus officinalis Sieb. et Zucc.*, *Lcycium barbarumL.*, *Prunus mume (Sieb.) Sieb.etZucc.*, *Paeonia lactiflora Pall.*, *Amomum villosum Lour.*, *Alpinia oxyphylla Miq.*, *Boehmeria nivea(L.)Gaud.*, *Scutellaria baicalensis Georgi*, *Artemisia argyi Levl.et Vant.* [12–14]. The conventional therapy should remain the same in the control group in the same RCT. If trials included other interventions such as acupuncture, acupoint application, and moxibustion, they were excluded.

*Control interventions.* The control group included studies using conventional therapy. conventional therapy include standard supportive care, human chorionic gonadotropin, progestogen drugs and other placebo. Standard supportive care include progestogens, iron, folic acid and multivitamin supplements and bed rest. The conventional therapy should remain the same in the experimental interventions in the same RCT.

## Exclusion criteria

Semi-randomized controlled trials, retrospective studies experience summaries, case series, case reports, conference reports, animal experiments and reviews.

## Outcomes

**Primary outcomes.** The primary outcome measure the number of miscarriage (defined as the absence of a fetal heartbeat on ultrasonography or spontaneous loss of pregnancy until week 12 of gestation). All the treatments will be performed from onset of threatened miscarriage symptoms until week 12 of pregnancy.

**Secondary outcomes.** Secondary outcomes are as follows: (1)the total effective rate (defined as the birth of a live-born baby until delivery), (2) Human chorionic gonadotropin (mIU/ml), (3) progesterone (ng/ml), (4) estradiol (pmol/L), (5) the shorten time of abdominal pain(the time from pain to no pain after drug treatment), (6) the shorten time of vaginal bleeding(the time from bleeding to zero bleeding after drug treatment), (7) the shorten time of lumbago(the time from pain to no pain after drug treatment), (8) adverse reactions.

## Information source and search strategy

The systematic review will include all randomized controlled trials (RCTs) studies published until April 2021. Electronic sources including CNKI, WF, VIP, CBM, MEDLINE(PubMed), Embase, Cochrane Library, and Web of Science will be searched for potentially eligible studies. The international clinical trial registration platform clinicaltrial.gov and the Chinese clinical trial registration platform of controlled trials will be searched from their inception. The search

**Table 2. Search strategy used in PubMed database.**

| Number | Search terms |
|---|---|
| 1 | threatened miscarriage[mh] |
| 2 | miscarriage[mh] |
| 3 | threatened abortion[mh] |
| 4 | abortion[mh] |
| 5 | fetal loss[mh] |
| 6 | pregnancy loss[mh] |
| 7 | or 1–6 |
| 8 | Yunkang oral liquid [mh] |
| 9 | randomized controlled trial [pt] |
| 10 | controlled clinical trial [pt] |
| 11 | randomized [tiab] |
| 12 | placebo [tiab] |
| 13 | drug therapy [sh] |
| 14 | randomly [tiab] |
| 15 | trial [tiab] |
| 16 | groups [tiab] |
| 17 | or 9–16 |
| 18 | humans[mh] |
| 19 | animal[mh] |
| 20 | 18 not 19 |
| 21 | 7 and 8 and 17 and 20 |

strategy will be based on the guidance of the Cochrane handbook. The search strategy for PubMed is shown in Table 2.

## Data screening

Two investigators will independently screen and cross-check the include studies. The studies obtained in the search will be imported into database for deduplication. Titles and abstracts will be read for preliminary screening of irrelevant studies and records, and the full papers will be read for excluding of non-randomized controlled trial, ineligible patients, ineligible controls, repeat published studies to determine included studies. When there are disagreements, three researchers will resolve the disagreements. Details of the selection procedure for studies are shown in a PRISMA-P flow chart (Fig 1).

## Data extraction and management

Two investigators will independently collect data from reports, and disagreement will be resolved by three investigators through consultation. We will extract the following data from the included articles: first author's name, time of publication, sample size, patients age, obstetric and gynecological history, gestational ages, the dosage of YKOL, the type of YKOL, the duration of treatment, the treatment method of control group, the number of miscarriage, adverse effects.

## Risk of bias assessment

The risk of bias of the included RCTs was assessed using the Cochrane 5.1.0 as-sessment tool. Evaluation items are as follows: the random sequence generation, concealment of allocation,

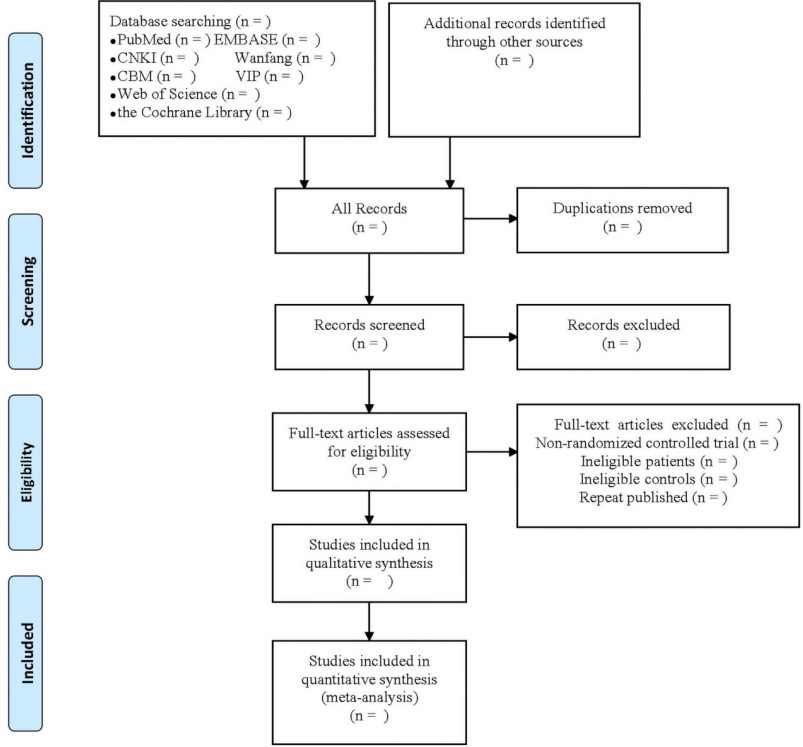

**Fig 1. PRISMA-P flow diagram.**

blinding of participants and personnel, blinding of outcomes assessments, incomplete outcome data, selective reporting, and other biases. Summary of each item results with a high, low, or unclear risk of bias will be displayed as a table. Two investigators will independently assess the quality of the included studies, and disagreement in risk of bias will be resolved by three investigators through consultation.

## Statistical analysis

**Data analysis and processing.** RevMan 5.4.1 will be used for data analysis. Odds ratio (OR) or relative risk (RR), 95% Confidence interval (CI) and P values will be used to estimate dichotomous variables include. The continuous data will be analyzed by Mean difference (MD) or Std Mean difference (SMD), 95% Confidence interval (CI) and P values. Concurrently, The Q value test and $I^2$ index was used to measure the statistical heterogeneity. When the heterogeneity exists ($I^2 > 50\%$ or $P < 0.1$), random-effects model was applied to estimate the summary of RR, WMD and 95% CI, otherwise a fixed-effects model for meta-analysis was used [21]. The meta-analyses will be based on the random effects model.

**Dealing with missing data.** If there is missing data in the RCT, the researchers will contact the corresponding author by email. If the data is still not available, the study will be excluded by two researchers through consultation.

**Subgroup analysis.** If the Q value test and $I^2$ indicate statistical heterogeneity, the source of heterogeneity was explored by analyzing the variables in two preset subgroups, include therapy time (one week as the cut-off point), history of abortion(an abortion as the cut-off point).

**Sensitivity analysis.** Sensitivity analysis will be performed by excluding studies with high risk of bias and changing the statistical model.

**Publication bias.** When ten or more studies are included, we will use a funnel plot to assess publication bias. However, due to the limitations of a funnel plot, egger's test will be used to help assess publication bias.

**Ethics and dissemination.** Ethical approval is not required for this article, our plan will be published in the journal. The results may be published in a peer-reviewed journal or disseminated in relevant conferences.

**Quality of evidence.** The included literature evaluates the evidence in the light of GRADE method, levels of quality of evidence were ranked as 4 levels: high, moderate, low, and very low. The quality of evidence was downgraded according to five domains: (I) limitation of the study design (if most domains had unclear bias risk, the evidence was rated down by one level. If poor trials were present and the results had poor robustness, the evidence was rated down by two levels), (II) Inconsistency (statistical heterogeneity with poor robustness after removing the trials with underestimated ADRs or overestimated efficacy), (III) indirection (the patients, interventions, outcomes or comparison of the study did not meet the objectives of this study), (IV) imprecision (the number of events for each evidence was less than 300), (V) publication bias (reporting bias and the results with poor robustness). Except for domain I, the evidence for domains II-IV was rated down by one level [21]. All processes are on this page: https://gradepro.org/.

## Supporting information

**S1 Table. PRISMA checklist.**
(DOC)

**S1 File. Individual search strategies.**
(DOCX)

**S2 File. Table of composition statement.**
(DOCX)

## Author Contributions

**Conceptualization:** Lingxia Xu, Fei Wang.

**Data curation:** Peng Sun, Liping Tang, Dongmei Yan.

**Formal analysis:** Peng Sun, Liping Tang, Dongmei Yan.

**Funding acquisition:** Peng Sun, Dongmei Yan, Fei Wang.

**Investigation:** Peng Sun, Liping Tang, Dongmei Yan.

**Methodology:** Peng Sun, Liping Tang.

**Project administration:** Lingxia Xu, Fei Wang.

**Supervision:** Bin li.

**Validation:** Bin li, Lingxia Xu.

**Writing – original draft:** Peng Sun, Lingxia Xu.

**Writing – review & editing:** Lingxia Xu, Fei Wang.

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
