## [Decision Letter · Decision Letter 0]

24 Sep 2021

PONE-D-21-15902Efficacy and safety of Yunkang oral liquil combined with conventional therapy for threatened miscarriage of first-trimester pregnancy A protocol for systematic review and meta-analysisPLOS ONE

Dear Dr. xu,

Thank you for submitting your manuscript to PLOS ONE. After careful consideration, we feel that it has merit but does not fully meet PLOS ONE’s publication criteria as it currently stands. Therefore, we invite you to submit a revised version of the manuscript that addresses the points raised during the review process.

We look forward to receiving your revised manuscript.

Kind regards,

Federico Ferrari

Academic Editor

PLOS ONE

Journal Requirements:

4. We suggest you thoroughly copyedit your manuscript for language usage, spelling, and grammar. If you do not know anyone who can help you do this, you may wish to consider employing a professional scientific editing service.

Additional Editor Comments (if provided):

Please take care about the comment of the Reviewer #2 and the following: in particular, you provide a description of the contents, but not the actual formulation, dosage and so on. This information is crucial for readers outside of China to deeply understand this study and also to be able to repeat it elsewhere. Please, give a full description of YKOL regarding doses and mode of use in methods.

Failing to provide adequate and detailed answers may impact in the editorial decision. 

Reviewers' comments:

Reviewer's Responses to Questions

**Comments to the Author**

1. Does the manuscript provide a valid rationale for the proposed study, with clearly identified and justified research questions?

Reviewer #1: Partly

Reviewer #2: Yes

2. Is the protocol technically sound and planned in a manner that will lead to a meaningful outcome and allow testing the stated hypotheses?

Reviewer #1: No

Reviewer #2: Yes

3. Is the methodology feasible and described in sufficient detail to allow the work to be replicable?

Reviewer #1: No

Reviewer #2: Yes

4. Have the authors described where all data underlying the findings will be made available when the study is complete?

Reviewer #1: No

Reviewer #2: Yes

5. Is the manuscript presented in an intelligible fashion and written in standard English?

Reviewer #1: No

Reviewer #2: No

6. Review Comments to the Author

You may also provide optional suggestions and comments to authors that they might find helpful in planning their study.

Reviewer #1: Please see uploaded document

Please see uploaded document

Please see uploaded document

Please see uploaded document

Reviewer #2: Minor language corrections should be made.

Line 47, insert a space between the two words "and" and "increase".

Line 49 and 50, correct spelling for "decidua"instead of "deoidua"

7. PLOS authors have the option to publish the peer review history of their article (what does this mean?). If published, this will include your full peer review and any attached files.

Reviewer #1: No

Reviewer #2: **Yes: **RICARDO BARINI

---

## [Author Response · Author response to Decision Letter 0]

5 Nov 2021

respond to specific reviewer comments in the box below.

Dear reviewer:

We are very grateful to your comments for the manuscript. According with your advice, we amended the relevant part in manuscript. Some of your questions were answered below.

PLOS ONE Efficacy and safety of Yunkang oral liquid combined with conventional therapy for threatened miscarriage of first-trimester pregnancy A protocol for systematic review and meta-analysis

Methodological peer review Mia Schmidt-Hansen 2/9/21

Abstract:

There are a number of language issues, e.g., 

-“relevant randomized controlled trials (RCTs) were searched”. It is not the RCTs that are searched, but the databases. 

-Past tense used in abstract in places. This needs to be changed to future tense. 

Response: Thank you for the comments. The systematic review will include all randomized controlled trials (RCTs) published. And future tense used in abstract in places. We have corrected it in the revised manuscript. 

Search date given is in the past (April 2021). A protocol should be published before any review work takes place to allow the peer review comments of the protocol to be taken into account in the actual work.

Response: Thanks for your comments. Search date given is in the past (April 2021) with the aim of generating idea and determining include and exclude criterion. We will start search formally in December 2021. Furthermore, we will update search results again before systematic review and meta-analysis.

“The aim of this study was to assess the efficacy and safety of YKOL in the treatment of threatened miscarriage during the first-trimester pregnancy (TMFP).” Please expand on the aim taking clear that this is a systematic review not a single study.

Response: Thank you for the comments. 

The aim of this systematic review was to assess the efficacy and safety of YKOL in the treatment of threatened miscarriage during the first-trimester pregnancy (TMFP). We have corrected it in the revised manuscript.

What are the inclusion criteria for the RCTs? Please include details of the population, interventions, comparators and outcomes. 

Response: Thanks for your comments. 

The types of studies include parallel-design, cross-over design, but we only included the data at the end of the first stage for cross-over design. Case–control, cohort studies, Case series and studies without controls were excluded. 

Participants must be individuals diagnosed with threatened miscarriage, and woman with gestational age between 1 and 12 weeks (first-trimester pregnancy). Threatened miscarriage is diagnosed on the basis of documented fetal cardiac activity on ultrasound with a history of vaginal bleeding in the presence of a closed cervix. There will be no restrictions based on age, race/ethnicity, socioeconomic status, or geographic region.

The experimental interventions of included studies receive Chinese Patent Medicine(YKOL) as a basic formula, in combination with conventional therapy compared with conventional therapy, regardless of the dose, method of dosing, or duration of administration. The conventional therapy should remain the same in the control group in the same RCT. If trials included other interventions such as acupuncture, acupoint application, and moxibustion, they were excluded.

The control group of included studies receive conventional therapy. conventional therapy include standard supportive care, human chorionic gonadotropin (hCG), progestogen drugs and other placebo. Standard supportive care include progestogens, iron, folic acid and multivitamin supplements and bed rest. The conventional therapy should remain the same in the experimental interventions in the same RCT.

Outcomes include (1)the number of miscarriage (defined as the absence of a fetal heartbeat on ultrasonography or spontaneous loss of pregnancy until week 12 of gestation), (2) the total effective rate (defined as the birth of a live-born baby until delivery) , (3) Human chorionic gonadotropin (mIU/ml), (4) progesterone(ng/ml), (5) estradiol (pmol/L), (6) the shorten time of abdominal pain(the time from pain to no pain after drug treatment), (7) the shorten time of vaginal bleeding(the time from bleeding to zero bleeding after drug treatment), (8) the shorten time of lumbago(the time from pain to no pain after drug treatment), (9) adverse reactions.

What statistical analyses are you going to perform?

Response: Thank you for the comments. The statistical analysis include data analysis and processing, deal with missing data, subgroup analysis, sensitivity analysis, and publication bias. 

“After screening the literature based on the inclusion and exclusion criteria, the Recommendation, Assessment, Development, and Evaluation (GRADE) system will be used to evaluate the quality evidence of each result.” There are a number of steps missing between screening the literature and applying GRADE. Including the information requested above will go some what in addressing this gap, but the sentence need revising.

Response: Thanks for your comments. The strength of evidence from the studies will be evaluated with the Grading of Recommendation, Assessment, Development and Evaluation (GRADE) methods. We have corrected it in the revised manuscript.

“Systematic review registration: INPLASY202140105.” What does this number refer to? Where is it registered? Why are you registering it here too? Due to the fact that you’ll have to make changes as a result of peer review, the protocol records will not be the same for the same review, so I don’t understand why you are going for this protocol publication? 

Response: Thank you for the comments.

 This number refer to this systematic review protocol has been registered on INPLASY website and registration number were INPLASY202140105 (https://www.doi.org/10.37766/inplasy2021.4.0105).

This systematic review protocol has been registered on INPLASY website(https://inplasy.com/).

The previous registration websites of system review will provide a peer review for the submitted protocol, however, these websites no longer provide peer reviews due to the impact of the epidemic and the increased number of submitted protocols, so we want to publish this protocol in journal of PLOS ONE with the aim of improving the rigor of this protocol. In addition, the information about the protocol is limited by entry and layout in the registration website. We can provide a complete version of the protocol by publishing it in PLOS ONE. With papers published in official journals, we would be easier to obtain support funds for follow-up research.

Background:

What does “TMFP”, “YKOL” and “TKOL” mean?

Response: Thank you for the comments. TMFP mean threatened miscarriage during the first-trimester pregnancy, YKOL mean Yunkang oral liquid, TKOL is a typo. 

“However, the systematic reviews examining the efficacy and safety of YKOL in the treatment of TMFP have never been systematically evaluated.” This sentence contradicts itself. Has systematic reviews been done or not?

Response: Thank you for the comments. Currently, the systematic reviews of YKOL have never been published.

Methods:

Please describe the types of included studies more. Do they have to be parallel-design or can they be cross-over?

Response: Thanks for your comments. The types of studies include parallel-design, cross-over design, but we only included the data at the end of the first stage for cross-over design. Case–control, cohort studies, Case series and studies without controls were excluded.

Please describe the participants more. What are the criteria for threatened miscarriage? You can reference the criteria you refer to, but they also need to be explicitly described. As an aside, I wonder why the women have to be 18 years? What if there is an otherwise eligible trial that has included women from 16 years? I recommend deleting the age criterion. You also need to describe the gestational age limits for inclusion.

Response: Thank you for the comments. Participants must be individuals diagnosed with threatened miscarriage, and woman with gestational age between 1 and 12 weeks (first-trimester pregnancy). Threatened miscarriage is diagnosed on the basis of documented fetal cardiac activity on ultrasound with a history of vaginal bleeding in the presence of a closed cervix. There will be no restrictions based on age, race/ethnicity, socioeconomic status, or geographic region.

The intervention needs to be much more described also, including how it is taken, how often, how much etc. What is conventional therapy? This also needs to be much more and better described. Are you effectively proposing to compare YKOL + standard care versus stand care alone? 

Is the comparator effectively standard care or other active drugs? What about placebo?

Response: Thanks for your comments.

The experimental interventions of included studies using Chinese Patent Medicine(YKOL) as a basic formula, in combination with conventional therapy compared with conventional therapy, regardless of the dose, method of dosing, or duration of administration. The conventional therapy should remain the same in the control group in the same RCT. If trials included other interventions such as acupuncture, acupoint application, and moxibustion, they were excluded.

Conventional therapy include standard supportive care, human chorionic gonadotropin (hCG), progestogen drugs and other placebo. Standard supportive care include progestogens, iron, folic acid and multivitamin supplements and bed rest.

When the included studies included YKOL + standard care versus standard care, YKOL + other active drugs versus other active drugs and YKOL versus placebo, it will be effectively compared.

Exclusion criteria: The test here needs extensive revision. It is a given that “1. The unrelated documents” will not be included, but “studies repeated publication” should not be deleted, but all publications of the same study should be used for data extraction, while of course treating the study as only one study”. It is also a given that “2. Semi-randomized controlled trials, retrospective studies experience summaries, case series, case reports, animal experiments and NARRATIVE reviews” will not be included. However, other systematic reviews should be checked for missed studies. Point 3 (“3. Review articles without incomplete data and have obvious errors.”) does not make sense, firstly because review articles do not contain original data and also secondly (and much less relevant as the sentence needs to be deleted) “without incomplete data” must be a typo?

Response: Thank you for the comments. Semi-randomized controlled trials, retrospective studies experience summaries, case series, case reports, conference reports, animal experiments and reviews, other systematic reviews. We have corrected it in the revised manuscript. 

Primary outcomes: Please describe in more detail. How will it be measured and what time points relative to the intervention are you interested in?

Response: Thanks for your comments.

The primary outcome measures the number of miscarriage (defined as the absence of a fetal heartbeat on ultrasonography or spontaneous loss of pregnancy until week 12 of gestation), All the treatments will be performed from onset of threatened miscarriage symptoms until week 12 of pregnancy. We have corrected it in the revised manuscript.

Secondary outcomes: I don’t know what they mean as outcome measures? The same comment as for the primary outcome applies here.

Response: Thank you for the comments. 

(1)the total effective rate (defined as the birth of a live-born baby until delivery), (2) Human chorionic gonadotropin(mIU/ml), (3) progesterone (ng/ml), (4) estradiol (pmol/L), (5) the shorten time of abdominal pain(the time from pain to no pain after drug treatment), (6) the shorten time of vaginal bleeding(the time from bleeding to zero bleeding after drug treatment), (7) the shorten time of lumbago(the time from pain to no pain after drug treatment), (8) adverse reactions. We have corrected it in the revised manuscript.

Search strategy: Same comments apply as on the abstract. Why is the end date 1/4/21? Why are you limiting the language to English and Chinese? Ideally, there should be no language or publication status limits, but if there are, a rationale for such limits need to included.

Response: Thanks for your comments. Search date given is in the past (April 2021) with the aim of generating idea and determining include and exclude criterion. We will start search formally in December 2021. Furthermore, we will update search results again before systematic review and meta-analysis. And we have deleted the limitations of language.

Why are you not searching trials databases and recent conference proceedings? You also need to check any relevant systematic reviews for missed studies, and explicitly state that you will do this.

Response: Thank you for the comments. The systematic review will include all randomized controlled trials (RCTs) studies published until April 2021. Electronic sources including CNKI, WF, VIP, CBM, MEDLINE(PubMed), Embase, Cochrane Library, and Web of Science will be searched for potentially eligible studies. The international clinical trial registration platform clinicaltrial.gov and the Chinese clinical trial registration platform of controlled trials will be searched from their inception until April 2021. We have corrected it in the revised manuscript. 

Conference proceedings and relevant systematic reviews are included in the database. 

My expertise is not in searching, but line looks wrong of your search strategy (i.e., include animals not humans). Also, all the individual search strategies for each database need to be included in an appendix.

Response: Thanks for your comments. We have corrected it in the revised manuscript. all the individual search strategies for each database were included in an appendix.

“Data screening Two independent researchers will screen the eligible studies according to the eligibility criteria and exclusion criteria, and disagreement in risk of bias will be resolved by three investigators through consultation. When there are disagreements, three researchers will resolve the disagreements. Details of the selection procedure for studies are shown in a PRISMA-P flow chart (Fig. 1)” There are a number of steps missing in this description to do with sifting the search on title and abstract and then full text. What is the relevance to risk of bias here? Please see a published Cochrane intervention review protocol and the Cochrane Intervention review Handbook for further guidance. 

Response: Thank you for the comments. 

Two investigators will independently screen and cross-check the included studies. The studies obtained in the search will be imported into database for deduplication. Titles and abstracts will be read for preliminary screening of irrelevant studies and records, and the full papers will be read for excluding of non-randomized controlled trial, ineligible patients, ineligible controls, repeat published studies to determine included studies. We have corrected it in the revised manuscript. 

Risk of bias is typo here.

Data extraction and management: The choice of words here make it difficult to understand. What is “filtered” articles? What is “ultimate eligible studies”? In terms of the data that will be extracted, then this also needs to be revised and clarified, but what is the relevance of the first author’s country? Surely the country the study was conducted in is the relevant country to extract. What does diagnostic criteria, treatment period, time of pregnancy, basic therapies; treatment and control measurements: outcomes: baseline and follow-up data mean in this regard? Be explicit and clear.

Response: Thanks for your comments. ”filtered” articles defined as eligible studies in qualitative synthesis(meta-analysis). ultimate eligible studies defined as eligible studies in qualitative synthesis(meta-analysis) . We deleted incomprehensible words.

I have deleted the first author’s country. 

diagnostic criteria means eligibility criteria, include participants must be individuals diagnosed with threatened miscarriage, and woman with gestational age between 1 and 12 weeks (first-trimester pregnancy). There will be no restrictions based on age, race/ethnicity, socioeconomic status, or geographic region.

treatment period mean duration of intervention.

time of pregnancy mean gestational age.

basic therapies mean conventional therapy, include standard supportive care, human chorionic gonadotropin, progestogen drugs and other placebo.

treatment and control measurements mean treatment method of intervention group and treatment method of control group.

outcomes: baseline and follow-up data is a typo. We have corrected it.

Two investigators will independently collect data from reports, and disagreement will be resolved by three investigators through consultation. We will extract the following data from the included articles: first author's name, time of publication, sample size, patients age, obstetric and gynecological history, gestational ages, the dosage of YKOL, the type of YKOL, the duration of treatment, the measure of control group, the number of miscarriage, adverse effects. We have corrected it in the revised manuscript. 

Quality assessment: You need to state which checklist you will use to assess risk of bias. Review Manager is not a checklist. In your description of the checklist you also need to use the same words that it uses rather than for example “whether distribution is hidden”. You also need to explicitly detail which items will be assessed at the study level and which will be assessed at the outcome level. 

Response: Thank you for the comments. 

the risk of bias of the included RCTs was assessed using the Cochrane 5.1.0 as-sessment tool.

The random sequence generation, concealment of allocation, blinding of participants and personnel were assesses at the study level. blinding of outcomes assessments, incomplete outcome data, selective reporting were assessed at the outcome level, and other biases were assessed. We have corrected it in the revised manuscript. 

Statistical analysis: What will determine whether you use odds ratios or relative risks? What will determine whether you analyse mean differences or standardised mean differences? What will you do if the Q value test and I2 indicate statistical heterogeneity? What criteria will you use to determine whether there is heterogeneity? Why are you planning to use a random effects model and not fixed effects?

Response: Thanks for your comments. 

When the event rate is rare, OR and RR will be similar; When the event rate is common, OR and RR will differ. For Dichotomous Data, OR is difficult to understand, OR are used in case-control studies and can account for covariates, RR are used in cohort study and randomized controlled study.

Mean difference(MD) is the difference between the two means, with same unit to the original measurement of experiments. It truly reflects the experimental effect and eliminates the influence of the absolute value on the result. Standard Mean difference(SMD) can be simply understood as the quotient of the difference between the two means and the combined standard deviation. It not only eliminates the influence of the absolute value of some research, but also eliminates the influence of the measurement unit on the result. Therefore, we analyse the MD when the unit of experiments’ results are same, otherwise, SMD were analysed.

If the Q value test and I2 indicate statistical heterogeneity, the source of heterogeneity was explored by analyzing the variables in two preset subgroups, include therapy time (one week as the cut-off point), history of abortion(an abortion as the cut-off point).

where statistically significant heterogeneity was defined as I2 >50% or P < 0.10 in the Q test.

When the heterogeneity exists (I2 > 50% or P < 0.1), random-effects model was applied to estimate the summary of RR, WMD and 95% CI, otherwise a fixed-effects model for meta-analysis was used. And random effects meta-analysis can encompass heterogeneity. So we plan to use a random effects model.

Dealing with missing data: What is “elusive” data?

Response: Thank you for the comments. ”elusive” is a typo here. We deleted it. 

Subgroup analysis: What is “substantial heterogeneity”? What does “the grouping factor for Subgroup analysis will be executed according to therapy time and number of abortions” mean? 

Response: Thank you for the comments.. 

Substantial heterogeneity is a typo here. 

 If the Q value test and I2 indicate statistical heterogeneity, the source of heterogeneity was explored by analyzing the variables in two preset subgroups, include therapy time (one week as the cut-off point), history of abortion(an abortion as the cut-off point).

Sensitivity analysis: How will you “change the statistical model”? 

Response: Thanks for your comments. change the statistical model defined as change effect measure, for example, random-effect models and fixed-effect models change each other. We want to change effect measure with the aim of performing sensitivity analysis. 

Quality of evidence: How will you apply GRADE to your data? You need to detail what will make you downgrade 1 or 2 levels for each of the GRADE domains. 

Response: Thank you for the comments. 

The quality of evidence was downgraded according to five domains: (I) limitation of the study design (if most domains had unclear bias risk, the evidence was rated down by one level. If poor trials were present and the results had poor robustness, the evidence was rated down by two levels), (II) Inconsistency (statistical heterogeneity with poor robustness after removing the trials with underestimated ADRs or overestimated efficacy), (III) indirection (the patients, interventions, outcomes or comparison of the study did not meet the objectives of this study), (IV) imprecision (the number of events for each evidence was less than 300), (V) publication bias (reporting bias and the results with poor robustness). Except for domain I, the evidence for domains II-IV was rated down by one level.

A protocol should not include a discussion. Please delete.

Response: Thank you for the comments. We have deleted it.

respond to editor comments in the box below.

Response: We have corrected it in the revised manuscript. manuscript meets PLOS ONE's style requirements.

Response: This work is supported by National Natural Science Foundation of China (No. 81202911 and 81360509), Evidence based ability improvement project of traditional Chinese Medicine in The State Administration of Traditional Chinese Medicine of China (No.2019XZZX-LG005), Academician Workstation of Jiangxi University of Traditional Chinese Medicine Project (No.ysgzz201806) and First-class Discipline Construction Funding of Jiangxi Province (No.JXSYLXK-ZHYAO148).\\

Response: our manuscript have no data.

4. We suggest you thoroughly copyedit your manuscript for language usage, spelling, and grammar. If you do not know anyone who can help you do this, you may wish to consider employing a professional scientific editing service.

Response:we have copyedited our manuscript for language usage, spelling, and grammar.

---

## [Decision Letter · Decision Letter 1]

29 Nov 2021

PONE-D-21-15902R1Efficacy and safety of Yunkang oral liquid combined with conventional therapy for threatened miscarriage of first-trimester pregnancy A protocol for systematic review and meta-analysisPLOS ONE

Dear Dr. xu,

Thank you for submitting your manuscript to PLOS ONE. After careful consideration, we feel that it has merit but does not fully meet PLOS ONE’s publication criteria as it currently stands. Therefore, we invite you to submit a revised version of the manuscript that addresses the points raised during the review process.

Please provide exactly what the reviewer request, this is crucial to avoid reject of the paper. Please submit your revised manuscript by Jan 13 2022 11:59PM. If you will need more time than this to complete your revisions, please reply to this message or contact the journal office at plosone@plos.org. Please include the following items when submitting your revised manuscript:A rebuttal letter that responds to each point raised by the academic editor and reviewer(s). You should upload this letter as a separate file labeled 'Response to Reviewers'.A marked-up copy of your manuscript that highlights changes made to the original version. You should upload this as a separate file labeled 'Revised Manuscript with Track Changes'.An unmarked version of your revised paper without tracked changes. You should upload this as a separate file labeled 'Manuscript'.

We look forward to receiving your revised manuscript.

Kind regards,

Federico Ferrari

Academic Editor

PLOS ONE

Reviewers' comments:

Reviewer's Responses to Questions

**Comments to the Author**

1. Does the manuscript provide a valid rationale for the proposed study, with clearly identified and justified research questions?

Reviewer #2: Yes

2. Is the protocol technically sound and planned in a manner that will lead to a meaningful outcome and allow testing the stated hypotheses?

Reviewer #2: Partly

3. Is the methodology feasible and described in sufficient detail to allow the work to be replicable?

Reviewer #2: Yes

4. Have the authors described where all data underlying the findings will be made available when the study is complete?

Reviewer #2: Yes

5. Is the manuscript presented in an intelligible fashion and written in standard English?

Reviewer #2: Yes

6. Review Comments to the Author

You may also provide optional suggestions and comments to authors that they might find helpful in planning their study.

Reviewer #2: I have asked authors to provide a thorough description of the contentes and proportion used to prepare Yunkang oral preparation. They have added the contents, but no proportions. This is crucial for the replication of the study any one would desire to pursue.

7. PLOS authors have the option to publish the peer review history of their article (what does this mean?). If published, this will include your full peer review and any attached files.

Reviewer #2: **Yes: **RICARDO BARINI

---

## [Author Response · Author response to Decision Letter 1]

16 Dec 2021

Reviewer #2: I have asked authors to provide a thorough description of the contentes and proportion used to prepare Yunkang oral preparation. They have added the contents, but no proportions. This is crucial for the replication of the study any one would desire to pursue.

Response:Thank you for the comments. 

TABLE 1. Basic information of YKOL

Chinese botanical drugs Latin name Part of botanical drugs Proportion

Shan yao Dioscorea opposita Thunb. rhizome 125g

Xu duan Dipsacus asper Wall. Ex Henry root 75g

Huang qi Astragalus membranaceus（Fisch.）Bge. root 100g

Dang gui Angelica sinensis（Oliv.）Diels root 75g

Gou ji(qumao) Cibotium barometz（L.）J.Sm. rhizome 100g

Tu si zi(bing) Cuscuta australis R.Br. seed 75g

Sang ji sheng Taxillus chinensis（DC.）Danser stalk 50g

Du zhong(chao) Eucommia ulmoides Oliv. bark 75g

Bu gu zhi Psoralea corylifoliaL. fruit 75g

Dang shen Codonopsis pilosula (Franch.)Nannf. root 75g

Fu ling Poria cocos（Schw.）Wolf sclerotium 100g

Bai zhu(jiao) Atractylodes macrocephala Koidz. rhizome 75g

e’jiao Equus asinus L. skin 25g

Di huang Rehmannia glutinosa (Gaertn.) DC. rhizome 100g

Shan zhu yu Cornus officinalis Sieb. et Zucc. fruit 75g

Gou qi zi Lcycium barbarumL. fruit 100g

Wu mei Prunus mume（Sieb.）Sieb.etZucc. fruit 50g

Bai shao Paeonia lactiflora Pall. root 75g 

Sha ren Amomum villosum Lour. fruit 50g

Yi zhi Alpinia oxyphylla Miq. fruit 50g

Zhu ma gen Boehmeria nivea(L.)Gaud. root 75g

Huang qin Scutellaria baicalensis Georgi root 50g

Ai ye Artemisia argyi Levl.et Vant. foliage 8.3g

---

## [Editor Report · Decision Letter 2]

14 Jan 2022

PONE-D-21-15902R2Efficacy and safety of Yunkang oral liquid combined with conventional therapy for threatened miscarriage of first-trimester pregnancy :A protocol for systematic review and meta-analysisPLOS ONE

Dear Dr. xu,

Thank you for submitting your manuscript to PLOS ONE. After careful consideration, we feel that it has merit but does not fully meet PLOS ONE’s publication criteria as it currently stands. Therefore, we invite you to submit a revised version of the manuscript that addresses the points raised during the review process.

We look forward to receiving your revised manuscript.

Kind regards,

Federico Ferrari

Academic Editor

PLOS ONE

Journal Requirements:

Additional Editor Comments:

Please consider for each article if they specify the composition of Yunkang and modalities of administration, a sort of "composition statement".

If none of the articles specified the composition of Yunkang you have to add a paragraph in the article affirming that all these studies are biased given the lack of these crucial information and hence to reduce the strenght of your conclusion.

Further, I strongly suggest to provide a table with each articles and a column that highlights if they are compliant or not with the "composition statement".

---

## [Author Response · Author response to Decision Letter 2]

16 Jan 2022

Please respond to specific reviewer and editor comments in the box below. To review those comments, click the View Decision Letter link.

Dear reviewer:

We are very grateful to your comments for the manuscript. According with your advice, we amended the relevant part in manuscript. Some of your questions were answered below.

PLOS ONE Efficacy and safety of Yunkang oral liquid combined with conventional therapy for threatened miscarriage of first-trimester pregnancy A protocol for systematic review and meta-analysis

Methodological peer review Mia Schmidt-Hansen 2/9/21

Abstract:

There are a number of language issues, e.g., 

-“relevant randomized controlled trials (RCTs) were searched”. It is not the RCTs that are searched, but the databases. 

-Past tense used in abstract in places. This needs to be changed to future tense. 

Response: Thank you for the comments. The systematic review will include all randomized controlled trials (RCTs) published. And future tense used in abstract in places. We have corrected it in the revised manuscript. 

Search date given is in the past (April 2021). A protocol should be published before any review work takes place to allow the peer review comments of the protocol to be taken into account in the actual work.

Response: Thanks for your comments. Search date given is in the past (April 2021) with the aim of generating idea and determining include and exclude criterion. We will start search formally in December 2021. Furthermore, we will update search results again before systematic review and meta-analysis.

“The aim of this study was to assess the efficacy and safety of YKOL in the treatment of threatened miscarriage during the first-trimester pregnancy (TMFP).” Please expand on the aim taking clear that this is a systematic review not a single study.

Response: Thank you for the comments. 

The aim of this systematic review was to assess the efficacy and safety of YKOL in the treatment of threatened miscarriage during the first-trimester pregnancy (TMFP). We have corrected it in the revised manuscript.

What are the inclusion criteria for the RCTs? Please include details of the population, interventions, comparators and outcomes. 

Response: Thanks for your comments. 

The types of studies include parallel-design, cross-over design, but we only included the data at the end of the first stage for cross-over design. Case–control, cohort studies, Case series and studies without controls were excluded. 

Participants must be individuals diagnosed with threatened miscarriage, and woman with gestational age between 1 and 12 weeks (first-trimester pregnancy). Threatened miscarriage is diagnosed on the basis of documented fetal cardiac activity on ultrasound with a history of vaginal bleeding in the presence of a closed cervix. There will be no restrictions based on age, race/ethnicity, socioeconomic status, or geographic region.

The experimental interventions of included studies receive Chinese Patent Medicine(YKOL) as a basic formula, in combination with conventional therapy compared with conventional therapy, regardless of the dose, method of dosing, or duration of administration. The conventional therapy should remain the same in the control group in the same RCT. If trials included other interventions such as acupuncture, acupoint application, and moxibustion, they were excluded.

The control group of included studies receive conventional therapy. conventional therapy include standard supportive care, human chorionic gonadotropin (hCG), progestogen drugs and other placebo. Standard supportive care include progestogens, iron, folic acid and multivitamin supplements and bed rest. The conventional therapy should remain the same in the experimental interventions in the same RCT.

Outcomes include (1)the number of miscarriage (defined as the absence of a fetal heartbeat on ultrasonography or spontaneous loss of pregnancy until week 12 of gestation), (2) the total effective rate (defined as the birth of a live-born baby until delivery) , (3) Human chorionic gonadotropin (mIU/ml), (4) progesterone(ng/ml), (5) estradiol (pmol/L), (6) the shorten time of abdominal pain(the time from pain to no pain after drug treatment), (7) the shorten time of vaginal bleeding(the time from bleeding to zero bleeding after drug treatment), (8) the shorten time of lumbago(the time from pain to no pain after drug treatment), (9) adverse reactions.

What statistical analyses are you going to perform?

Response: Thank you for the comments. The statistical analysis include data analysis and processing, deal with missing data, subgroup analysis, sensitivity analysis, and publication bias. 

“After screening the literature based on the inclusion and exclusion criteria, the Recommendation, Assessment, Development, and Evaluation (GRADE) system will be used to evaluate the quality evidence of each result.” There are a number of steps missing between screening the literature and applying GRADE. Including the information requested above will go some what in addressing this gap, but the sentence need revising.

Response: Thanks for your comments. The strength of evidence from the studies will be evaluated with the Grading of Recommendation, Assessment, Development and Evaluation (GRADE) methods. We have corrected it in the revised manuscript.

“Systematic review registration: INPLASY202140105.” What does this number refer to? Where is it registered? Why are you registering it here too? Due to the fact that you’ll have to make changes as a result of peer review, the protocol records will not be the same for the same review, so I don’t understand why you are going for this protocol publication? 

Response: Thank you for the comments.

 This number refer to this systematic review protocol has been registered on INPLASY website and registration number were INPLASY202140105 (https://www.doi.org/10.37766/inplasy2021.4.0105).

This systematic review protocol has been registered on INPLASY website(https://inplasy.com/).

The previous registration websites of system review will provide a peer review for the submitted protocol, however, these websites no longer provide peer reviews due to the impact of the epidemic and the increased number of submitted protocols, so we want to publish this protocol in journal of PLOS ONE with the aim of improving the rigor of this protocol. In addition, the information about the protocol is limited by entry and layout in the registration website. We can provide a complete version of the protocol by publishing it in PLOS ONE. With papers published in official journals, we would be easier to obtain support funds for follow-up research.

Background:

What does “TMFP”, “YKOL” and “TKOL” mean?

Response: Thank you for the comments. TMFP mean threatened miscarriage during the first-trimester pregnancy, YKOL mean Yunkang oral liquid, TKOL is a typo. 

“However, the systematic reviews examining the efficacy and safety of YKOL in the treatment of TMFP have never been systematically evaluated.” This sentence contradicts itself. Has systematic reviews been done or not?

Response: Thank you for the comments. Currently, the systematic reviews of YKOL have never been published.

Methods:

Please describe the types of included studies more. Do they have to be parallel-design or can they be cross-over?

Response: Thanks for your comments. The types of studies include parallel-design, cross-over design, but we only included the data at the end of the first stage for cross-over design. Case–control, cohort studies, Case series and studies without controls were excluded.

Please describe the participants more. What are the criteria for threatened miscarriage? You can reference the criteria you refer to, but they also need to be explicitly described. As an aside, I wonder why the women have to be 18 years? What if there is an otherwise eligible trial that has included women from 16 years? I recommend deleting the age criterion. You also need to describe the gestational age limits for inclusion.

Response: Thank you for the comments. Participants must be individuals diagnosed with threatened miscarriage, and woman with gestational age between 1 and 12 weeks (first-trimester pregnancy). Threatened miscarriage is diagnosed on the basis of documented fetal cardiac activity on ultrasound with a history of vaginal bleeding in the presence of a closed cervix. There will be no restrictions based on age, race/ethnicity, socioeconomic status, or geographic region.

The intervention needs to be much more described also, including how it is taken, how often, how much etc. What is conventional therapy? This also needs to be much more and better described. Are you effectively proposing to compare YKOL + standard care versus stand care alone? 

Is the comparator effectively standard care or other active drugs? What about placebo?

Response: Thanks for your comments.

The experimental interventions of included studies using Chinese Patent Medicine(YKOL) as a basic formula (Table 1), in combination with conventional therapy compared with conventional therapy, regardless of the dose, method of dosing, or duration of administration. The conventional therapy should remain the same in the control group in the same RCT. If trials included other interventions such as acupuncture, acupoint application, and moxibustion, they were excluded.

TABLE 1. Basic information of YKOL

Chinese botanical drugs Latin name Part of botanical drugs Proportion

Shan yao Dioscorea opposita Thunb. rhizome 125g

Xu duan Dipsacus asper Wall. Ex Henry root 75g

Huang qi Astragalus membranaceus（Fisch.）Bge. root 100g

Dang gui Angelica sinensis（Oliv.）Diels root 75g

Gou ji(qumao) Cibotium barometz（L.）J.Sm. rhizome 100g

Tu si zi(bing) Cuscuta australis R.Br. seed 75g

Sang ji sheng Taxillus chinensis（DC.）Danser stalk 50g

Du zhong(chao) Eucommia ulmoides Oliv. bark 75g

Bu gu zhi Psoralea corylifoliaL. fruit 75g

Dang shen Codonopsis pilosula (Franch.)Nannf. root 75g

Fu ling Poria cocos（Schw.）Wolf sclerotium 100g

Bai zhu(jiao) Atractylodes macrocephala Koidz. rhizome 75g

e’jiao Equus asinus L. skin 25g

Di huang Rehmannia glutinosa (Gaertn.) DC. rhizome 100g

Shan zhu yu Cornus officinalis Sieb. et Zucc. fruit 75g

Gou qi zi Lcycium barbarumL. fruit 100g

Wu mei Prunus mume（Sieb.）Sieb.etZucc. fruit 50g

Bai shao Paeonia lactiflora Pall. root 75g

Sha ren Amomum villosum Lour. fruit 50g

Yi zhi Alpinia oxyphylla Miq. fruit 50g

Zhu ma gen Boehmeria nivea(L.)Gaud. root 75g

Huang qin Scutellaria baicalensis Georgi root 50g

Ai ye Artemisia argyi Levl.et Vant. foliage 8.3g

Conventional therapy include standard supportive care, human chorionic gonadotropin (hCG), progestogen drugs and other placebo. Standard supportive care include progestogens, iron, folic acid and multivitamin supplements and bed rest.

When the included studies included YKOL + standard care versus standard care, YKOL + other active drugs versus other active drugs and YKOL versus placebo, it will be effectively compared.

Exclusion criteria: The test here needs extensive revision. It is a given that “1. The unrelated documents” will not be included, but “studies repeated publication” should not be deleted, but all publications of the same study should be used for data extraction, while of course treating the study as only one study”. It is also a given that “2. Semi-randomized controlled trials, retrospective studies experience summaries, case series, case reports, animal experiments and NARRATIVE reviews” will not be included. However, other systematic reviews should be checked for missed studies. Point 3 (“3. Review articles without incomplete data and have obvious errors.”) does not make sense, firstly because review articles do not contain original data and also secondly (and much less relevant as the sentence needs to be deleted) “without incomplete data” must be a typo?

Response: Thank you for the comments. Semi-randomized controlled trials, retrospective studies experience summaries, case series, case reports, conference reports, animal experiments and reviews, other systematic reviews. We have corrected it in the revised manuscript. 

Primary outcomes: Please describe in more detail. How will it be measured and what time points relative to the intervention are you interested in?

Response: Thanks for your comments.

The primary outcome measures the number of miscarriage (defined as the absence of a fetal heartbeat on ultrasonography or spontaneous loss of pregnancy until week 12 of gestation), All the treatments will be performed from onset of threatened miscarriage symptoms until week 12 of pregnancy. We have corrected it in the revised manuscript.

Secondary outcomes: I don’t know what they mean as outcome measures? The same comment as for the primary outcome applies here.

Response: Thank you for the comments. 

(1)the total effective rate (defined as the birth of a live-born baby until delivery), (2) Human chorionic gonadotropin (mIU/ml), (3) progesterone (ng/ml), (4) estradiol (pmol/L), (5) the shorten time of abdominal pain(the time from pain to no pain after drug treatment), (6) the shorten time of vaginal bleeding(the time from bleeding to zero bleeding after drug treatment), (7) the shorten time of lumbago(the time from pain to no pain after drug treatment), (8) adverse reactions. We have corrected it in the revised manuscript.

Search strategy: Same comments apply as on the abstract. Why is the end date 1/4/21? Why are you limiting the language to English and Chinese? Ideally, there should be no language or publication status limits, but if there are, a rationale for such limits need to included.

Response: Thanks for your comments. Search date given is in the past (April 2021) with the aim of generating idea and determining include and exclude criterion. We will start search formally in December 2021. Furthermore, we will update search results again before systematic review and meta-analysis. And we have deleted the limitations of language.

Why are you not searching trials databases and recent conference proceedings? You also need to check any relevant systematic reviews for missed studies, and explicitly state that you will do this.

Response: Thank you for the comments. The systematic review will include all randomized controlled trials (RCTs) studies published until April 2021. Electronic sources including CNKI, WF, VIP, CBM, MEDLINE(PubMed), Embase, Cochrane Library, and Web of Science will be searched for potentially eligible studies. The international clinical trial registration platform clinicaltrial.gov and the Chinese clinical trial registration platform of controlled trials will be searched from their inception until April 2021. We have corrected it in the revised manuscript. 

Conference proceedings and relevant systematic reviews are included in the database. 

My expertise is not in searching, but line looks wrong of your search strategy (i.e., include animals not humans). Also, all the individual search strategies for each database need to be included in an appendix.

Response: Thanks for your comments. We have corrected it in the revised manuscript. all the individual search strategies for each database were included in an appendix.

“Data screening Two independent researchers will screen the eligible studies according to the eligibility criteria and exclusion criteria, and disagreement in risk of bias will be resolved by three investigators through consultation. When there are disagreements, three researchers will resolve the disagreements. Details of the selection procedure for studies are shown in a PRISMA-P flow chart (Fig. 1)” There are a number of steps missing in this description to do with sifting the search on title and abstract and then full text. What is the relevance to risk of bias here? Please see a published Cochrane intervention review protocol and the Cochrane Intervention review Handbook for further guidance. 

Response: Thank you for the comments. 

Two investigators will independently screen and cross-check the included studies. The studies obtained in the search will be imported into database for deduplication. Titles and abstracts will be read for preliminary screening of irrelevant studies and records, and the full papers will be read for excluding of non-randomized controlled trial, ineligible patients, ineligible controls, repeat published studies to determine included studies. We have corrected it in the revised manuscript. 

Risk of bias is typo here.

Data extraction and management: The choice of words here make it difficult to understand. What is “filtered” articles? What is “ultimate eligible studies”? In terms of the data that will be extracted, then this also needs to be revised and clarified, but what is the relevance of the first author’s country? Surely the country the study was conducted in is the relevant country to extract. What does diagnostic criteria, treatment period, time of pregnancy, basic therapies; treatment and control measurements: outcomes: baseline and follow-up data mean in this regard? Be explicit and clear.

Response: Thanks for your comments. ”filtered” articles defined as eligible studies in qualitative synthesis(meta-analysis). ultimate eligible studies defined as eligible studies in qualitative synthesis(meta-analysis) . We deleted incomprehensible words.

I have deleted the first author’s country. 

diagnostic criteria means eligibility criteria, include participants must be individuals diagnosed with threatened miscarriage, and woman with gestational age between 1 and 12 weeks (first-trimester pregnancy). There will be no restrictions based on age, race/ethnicity, socioeconomic status, or geographic region.

treatment period mean duration of intervention.

time of pregnancy mean gestational age.

basic therapies mean conventional therapy, include standard supportive care, human chorionic gonadotropin, progestogen drugs and other placebo.

treatment and control measurements mean treatment method of intervention group and treatment method of control group.

outcomes: baseline and follow-up data is a typo. We have corrected it.

Two investigators will independently collect data from reports, and disagreement will be resolved by three investigators through consultation. We will extract the following data from the included articles: first author's name, time of publication, sample size, patients age, obstetric and gynecological history, gestational ages, the dosage of YKOL, the type of YKOL, the duration of treatment, the measure of control group, the number of miscarriage, adverse effects. We have corrected it in the revised manuscript. 

Quality assessment: You need to state which checklist you will use to assess risk of bias. Review Manager is not a checklist. In your description of the checklist you also need to use the same words that it uses rather than for example “whether distribution is hidden”. You also need to explicitly detail which items will be assessed at the study level and which will be assessed at the outcome level. 

Response: Thank you for the comments. 

the risk of bias of the included RCTs was assessed using the Cochrane 5.1.0 as-sessment tool.

The random sequence generation, concealment of allocation, blinding of participants and personnel were assesses at the study level. blinding of outcomes assessments, incomplete outcome data, selective reporting were assessed at the outcome level, and other biases were assessed. We have corrected it in the revised manuscript. 

Statistical analysis: What will determine whether you use odds ratios or relative risks? What will determine whether you analyse mean differences or standardised mean differences? What will you do if the Q value test and I2 indicate statistical heterogeneity? What criteria will you use to determine whether there is heterogeneity? Why are you planning to use a random effects model and not fixed effects?

Response: Thanks for your comments. 

When the event rate is rare, OR and RR will be similar; When the event rate is common, OR and RR will differ. For Dichotomous Data, OR is difficult to understand, OR are used in case-control studies and can account for covariates, RR are used in cohort study and randomized controlled study.

Mean difference(MD) is the difference between the two means, with same unit to the original measurement of experiments. It truly reflects the experimental effect and eliminates the influence of the absolute value on the result. Standard Mean difference(SMD) can be simply understood as the quotient of the difference between the two means and the combined standard deviation. It not only eliminates the influence of the absolute value of some research, but also eliminates the influence of the measurement unit on the result. Therefore, we analyse the MD when the unit of experiments’ results are same, otherwise, SMD were analysed.

If the Q value test and I2 indicate statistical heterogeneity, the source of heterogeneity was explored by analyzing the variables in two preset subgroups, include therapy time (one week as the cut-off point), history of abortion(an abortion as the cut-off point).

where statistically significant heterogeneity was defined as I2 >50% or P < 0.10 in the Q test.

When the heterogeneity exists (I2 > 50% or P < 0.1), random-effects model was applied to estimate the summary of RR, WMD and 95% CI, otherwise a fixed-effects model for meta-analysis was used. And random effects meta-analysis can encompass heterogeneity. So we plan to use a random effects model.

Dealing with missing data: What is “elusive” data?

Response: Thank you for the comments. ”elusive” is a typo here. We deleted it. 

Subgroup analysis: What is “substantial heterogeneity”? What does “the grouping factor for Subgroup analysis will be executed according to therapy time and number of abortions” mean? 

Response: Thank you for the comments.. 

Substantial heterogeneity is a typo here. 

 If the Q value test and I2 indicate statistical heterogeneity, the source of heterogeneity was explored by analyzing the variables in two preset subgroups, include therapy time (one week as the cut-off point), history of abortion(an abortion as the cut-off point).

Sensitivity analysis: How will you “change the statistical model”? 

Response: Thanks for your comments. change the statistical model defined as change effect measure, for example, random-effect models and fixed-effect models change each other. We want to change effect measure with the aim of performing sensitivity analysis. 

Quality of evidence: How will you apply GRADE to your data? You need to detail what will make you downgrade 1 or 2 levels for each of the GRADE domains. 

Response: Thank you for the comments. 

The quality of evidence was downgraded according to five domains: (I) limitation of the study design (if most domains had unclear bias risk, the evidence was rated down by one level. If poor trials were present and the results had poor robustness, the evidence was rated down by two levels), (II) Inconsistency (statistical heterogeneity with poor robustness after removing the trials with underestimated ADRs or overestimated efficacy), (III) indirection (the patients, interventions, outcomes or comparison of the study did not meet the objectives of this study), (IV) imprecision (the number of events for each evidence was less than 300), (V) publication bias (reporting bias and the results with poor robustness). Except for domain I, the evidence for domains II-IV was rated down by one level.

A protocol should not include a discussion. Please delete.

Response: Thank you for the comments. We have deleted it.

Dear Editor:

RE: PONE-D-21-15902R2

We would like to thank PLOS ONE for giving us the opportunity to revise our manuscript. We thank the editors for their careful read and thoughtful comments on previous draft. We have carefully taken their comments into consideration in preparing our revision, which has resulted in a paper that is clearer, more compelling, and broader. The following summarizes how we responded to reviewer comments. Response to Reviewers file is our response to their comments. We have changed our financial disclosure(Evidence based ability improvement project of traditional Chinese Medicine in The State Administration of Traditional Chinese Medicine of China (No.2019XZZX-LG005)). In addition, we provide a table with each articles and a column that highlights if they are compliant or not with the "composition statement".

Yours sincerely,

Lingxia, Xu

On behalf of the research team

---

## [Editor Report · Decision Letter 3]

24 Jan 2022

Efficacy and safety of Yunkang oral liquid combined with conventional therapy for threatened miscarriage of first-trimester pregnancy :A protocol for systematic review and meta-analysis

PONE-D-21-15902R3

Dear Dr. xu,

We’re pleased to inform you that your manuscript has been judged scientifically suitable for publication and will be formally accepted for publication once it meets all outstanding technical requirements.

Kind regards,

Federico Ferrari

Academic Editor

PLOS ONE
---

## [Editor Report · Acceptance letter]

28 Jan 2022

PONE-D-21-15902R3 

Efficacy and safety of Yunkang oral liquid combined with conventional therapy for threatened miscarriage of first-trimester pregnancy A protocol for systematic review and meta-analysis 

Dear Dr. Xu:

I'm pleased to inform you that your manuscript has been deemed suitable for publication in PLOS ONE. Congratulations! Your manuscript is now with our production department. 

Kind regards, 

on behalf of

Dr. Federico Ferrari 

Academic Editor

PLOS ONE